# The Performance of Zr-Doped Al-Zn-Sn-O Thin Film Transistor Prepared by Co-Sputtering

**Xiaochen Zhang [1], Xianzhe Liu [1], Kuankuan Lu [1], Honglong Ning [1,*], Dong Guo [2], Yiping Wang [3], Zhihang Li [1], Muyang Shi [1], Rihui Yao [1,*] and Junbiao Peng [1]**

[1]    Institute of Polymer Optoelectronic Materials and Devices, State Key Laboratory of Luminescent Materials and Devices, South China University of Technology, Guangzhou 510640, China; zhangxc_scut@foxmail.com (X.Z.); msliuxianzhe@mail.scut.edu.cn (X.L.); mskk-lu@mail.scut.edu.cn (K.L.); mslzhscut@mail.scut.edu.cn (Z.L.); 201430320229@mail.scut.edu.cn (M.S.); psjbpeng@scut.edu.cn (J.P.)

[2]    School of Materials Science and Engineering, Beihang University, Beijing 100191, China; dong.guo@buaa.edu.cn

[3]    State Key Laboratory of Mechanics and Control of Mechanical Structures, Nanjing University of Aeronautics and Astronautics, Nanjing 210016, China; yipingwang@nuaa.edu.cn

*    Correspondence: ninghl@scut.edu.cn (H.N.); yaorihui@scut.edu.cn (R.Y.)

**Abstract:** In this work, a thin film transistor (TFT) with Zr-doped aluminum-zinc-tin oxide (Zr-AZTO) semiconductor as active layer was investigated. The Zr-AZTO thin films were co-sputtered by $ZrO_2$ and AZTO targets (RF-Sputter) in Ar, and annealed at 350 °C in air atmosphere. With the discharge power of AZTO increasing from 100 W to 120 W, the content of Zr element decreases from 0.63 ± 0.01 at.% to 0.34 ± 0.01 at.%, and the oxygen vacancy decreases from (19.0 ± 0.1)% to (17.3 ± 0.8)%. The results of Zr-AZTO thin film show that the main factor is the co-sputter power of $ZrO_2$ target. With the co-sputter power of $ZrO_2$ increasing from 15 W to 45 W, the content of Zr element increases from 0.63 ± 0.01 at.% to 2.79 ± 0.01 at.%, the content of oxygen vacancy decreases from (19.0 ± 0.1)% to (14.1 ± 0.1)%, Eg increases from 2.76 eV to 2.86 eV, and the root mean square (RMS) roughness firstly decreases from 0.402 nm to 0.387 nm and then increases to 0.490 nm. The Micro Wave Photo Conductivity Decay (μ-PCD, LTA-1620SP) was used to measure the quality of Zr-AZTO thin film and the mean peak and D value decreases from 139.3 mV to 80.9 mV and from 1.54 to 0.77 as the co-sputter power of $ZrO_2$ increases from 15 W to 45 W, which means it has highest localized states and defects in high $ZrO_2$ co-sputter power. The devices prepared at 15 W ($ZrO_2$)/100 W (AZTO) co-sputter show a best performance, with a $μ_{sat}$ of 8.0 ± 0.6 cm$^2$/(V·S), an Ion/Ioff of (2.01 ± 0.34) × 10$^6$, and a SS of 0.18 ± 0.03 V/dec. The device of Sample B has a 0.5 V negative shift under −20 V NBS and 9.6 V positive shift under 20 V PBS.

**Keywords:** thin film transistor; oxide semiconductor; Zr-AZTO; co-sputter

## 1. Introduction

The research on metal oxide semiconductors (MOS) has drawn researchers' attention for several years. Due to its various advantages: visible-light transparency, great uniformity, high mobility and lower processing temperature, MOS materials have been applied on an active matrix display field to replace amorphous silicon (a-Si) [1,2]. Amorphous indium-gallium-zinc oxide (a-IGZO) is the most conventional material [3,4], however, the low content in the earth and high cost of gallium and indium limit the application of a-IGZO [5,6]. Therefore, new MOS materials without indium have attracted researchers' interest [7–10], and a non-indium MOS material (AZTO etc.) is a new choice to solve the problem of a-IGZO [11,12].

Before this work, there were no reports about Zr-AZTO work as an active layer of a TFT device. In this paper, the Zr-doped AZTO (Zr-AZTO) thin film and TFT devices were prepared by co-sputtering, and how Zr element in the film influences the electrical properties of devices was investigated. X-ray photoelectron spectroscopy (XPS, Thermo ESCALAB 250 Xi) was used to ascertain the Zr content and the O 1 s peak which could represent the carrier concentration. With the discharge power of $ZrO_2$ increasing, the oxygen vacancy of Zr-AZTO films decreases, which means the discharge power of $ZrO_2$ target can adjust oxygen vacancy of Zr-AZTO films and the high concentration of oxygen vacancy was obtained with a low discharge power of $ZrO_2$. The XRD measurement (PANalytical Empyrean, Cu X-ray target) shows that the Zr-AZTO films under different conditions are amorphous, which means the Zr-AZTO can be used on large-scale display because of its good uniformity. The localized states in the active layer which could affect the charge carrier mobility of TFTs were evaluated by μ-PCD. The μ-PCD shows that the Zr-AZTO has low density of localized states when prepared under a low discharge power of the $ZrO_2$ target. With high concentration of oxygen vacancy, low localized states, and an amorphous structure, the Zr-AZTO has the potential to apply on TFT devices.

## 2. Materials and Methods

The TFT devices were fabricated on the glass substrate with a bottom-gate structure. First, 300-nm-thick Al-Nd alloy layer was deposited as gate electrode by direct-current (DC) magnetron sputtering and chemically patterned by wet etching. Secondly, a 200-nm-thick Nd-doped $AlO_x$ layer was covered on the gate electrode by anodic oxidation. The glass substrate containing 300 nm Al:Nd was immersed in a 3.48 wt.% ammonium tartrate solution (volume ratio of glycol to ammonium tartrate is 3:1). Al-Nd was connected with anode and Pt was connected with cathode. A fixed current of 0.1 mA/cm$^{-2}$ was applied and the voltage was increased linearly to 100 volts, then voltage was fixed at 100 volts and the current was decreased until it reached less than 1 μA. After the anodic oxidation, 200 nm in Al:Nd was oxidized into Nd-doped $AlO_x$. Then a 20-nm-thick Zr-AZTO layer was formed by vapor co-sputtering process with 10 cm target-substrate distance in a 5 mTorr mixed atmosphere (Ar:$O_2$ = 20:1) at room temperature. To fabricate devices with a different Zr content, the $ZrO_2$ target (ϕ50.8 *6 mm) was sputtered by 15 W, 25 W, 35 W, and 45 W while the AZTO target (ϕ50.8 *3 mm) with an $Al_2O_3$:ZnO:$SnO_2$ ratio of 2:23:75 wt.% was sputtered by 100 W and 120 W. After that, the devices were annealed at 350 °C in air atmosphere. Finally, the 150-nm-thick Al source and drain electrodes patterned by shadow mask were deposited by 80 W DC magnetron sputtering. With the shadow mask, the channel width and Length was controlled at 500 nm and 700 nm. The electrical characteristics of the devices were measured by a semiconductor parameter analyzer in an air atmosphere. The Zr-AZTO films with different Zr content were tested by μ-PCD to reveal the concentration of carrier in the film and the relationship between the carrier concentration and electrical performance of devices.

## 3. Results and Discussion

Table 1 shows the power combinations used to prepare the Zr-AZTO. The variate of Sample A and Sample B is the discharge power of AZTO, which is 120 W and 100 W. The variate of Sample B, C, D, and E is the discharge power of $ZrO_2$ target, which is 15 W, 25 W, 35 W, and 45 W.

**Table 1.** The discharge power of Zr-doped aluminum-zinc-tin oxide (Zr-AZTO) fabrication.

|  | Sample A | Sample B | Sample C | Sample D | Sample E |
|---|---|---|---|---|---|
| Power of $ZrO_2$ (W) | 15 | 15 | 25 | 35 | 45 |
| Power of AZTO (W) | 120 | 100 | 100 | 100 | 100 |

Figure 1 shows the Zr-AZTO XPS of Zr 3d, indicating that there are two peaks at 182 eV (Zr $3d_{5/2}$) and 185 eV (Zr $3d_{3/2}$), and the element zirconium is present in the form of $ZrO_2$ [13,14]. From Sample A to Sample E, the Zr content respectively is 0.34 ± 0.01 at.%, 0.63 ± 0.01 at.%, 1.00 ± 0.01 at.%,

1.78 ± 0.01 at.%, and 2.79 ± 0.01 at.%. With the discharge power of AZTO target decreasing, Zr content increases from 0.34 ± 0.01 at.% to 0.63 ± 0.01 at.%. Figure 2a shows the two peaks merging together because of the low Zr content of Sample A. With the discharge power of $ZrO_2$ target increasing, Zr content increases from 0.63 ± 0.01 at.% to 2.79 ± 0.01 at.%, and the discharge power of $ZrO_2$ is the main factor of thin film Zr content.

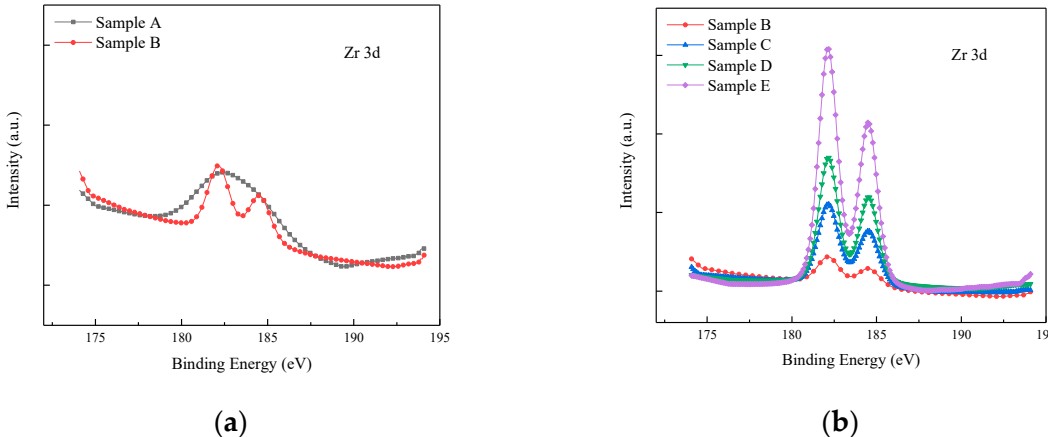

**Figure 1.** Zr-AZTO XPS of different co-sputter power: (**a**) 15 W $ZrO_2$ target; (**b**) 100 W AZTO target.

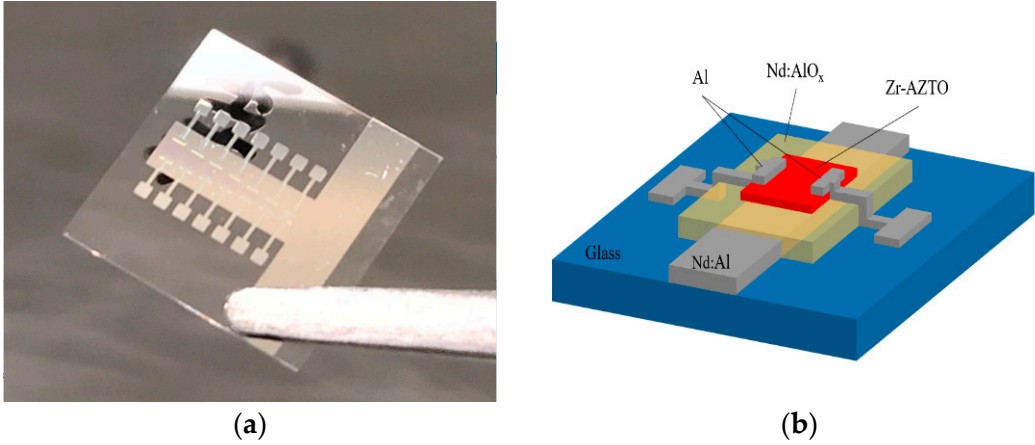

**Figure 2.** The image of practical device (**a**) and the schematic diagram of bottom gate Zr-AZTO TFT structure (**b**).

Table 2 is the peaks table of the elements used in Zr-AZTO films (except for Zr). For the Al element, the peak of Al $2p_{3/2}$ is around 73.60 eV, which indicates the Al element is present in the form of $Al_2O_3$ [15]. For Sn and Zn element, the peaks of Sn $3d_{5/2}$ and Zn $2p_{3/2}$ are about 486.60 eV and 1021.60 eV, which means the Sn and Zn element in the films are present in form of $SnO_2$ and ZnO [16,17].

**Table 2.** The peaks table of element in Zr-AZTO films.

|  | Sample A | Sample B | Sample C | Sample D | Sample E |
|---|---|---|---|---|---|
| Peak of Al $2p_{3/2}$ (eV) | 73.53 | 73.55 | 73.60 | 73.60 | 73.59 |
| Peak of Sn $3d_{5/2}$ (eV) | 486.60 | 486.60 | 486.60 | 486.58 | 486.59 |
| Peak of Zn $2p_{3/2}$ (eV) | 1021.61 | 1021.62 | 1021.63 | 1021.59 | 1021.59 |

Figure 3 is the O 1 s spectra of Zr-AZTO of different power combinations. The O 1 s peak can be divided into three sub-peaks: (1) the M-O-M peak at 530 eV represents the $O^{2-}$ binding with Zr,

Al, Sn, Zn; (2) the Vo peak at 531 eV represents the oxygen vacancy; (3) the M-O-R peak at 532 eV represents the weakly bond like $H_2O$, $-CO_3$ and $-OH$ [18]. The proportion of sub-peak area can indicate the different bond species content. Calculating with $S_{vo}/S_{all}$ ratio of Sample A and Sample B, it is found that the oxygen vacancy of Sample A~E is $17.3 \pm 0.8\%$, $19.0 \pm 0.1\%$, $16.0 \pm 0.7\%$, $15.4 \pm 0.6\%$, and $14.1 \pm 0.1\%$. With the discharge power of AZTO increasing from 100 W to 120 W, the oxygen vacancy decreases from $17.3 \pm 0.8\%$ to $19.0 \pm 0.1\%$ and with the discharge power of $ZrO_2$ increasing from 15 W to 45 W, the content of oxygen vacancy decreases from $19.0 \pm 0.1\%$ to $14.1 \pm 0.1\%$. Since it is known that the carriers of oxide semiconductors are related to oxygen vacancy, the Sample B has a highest carrier density [19].

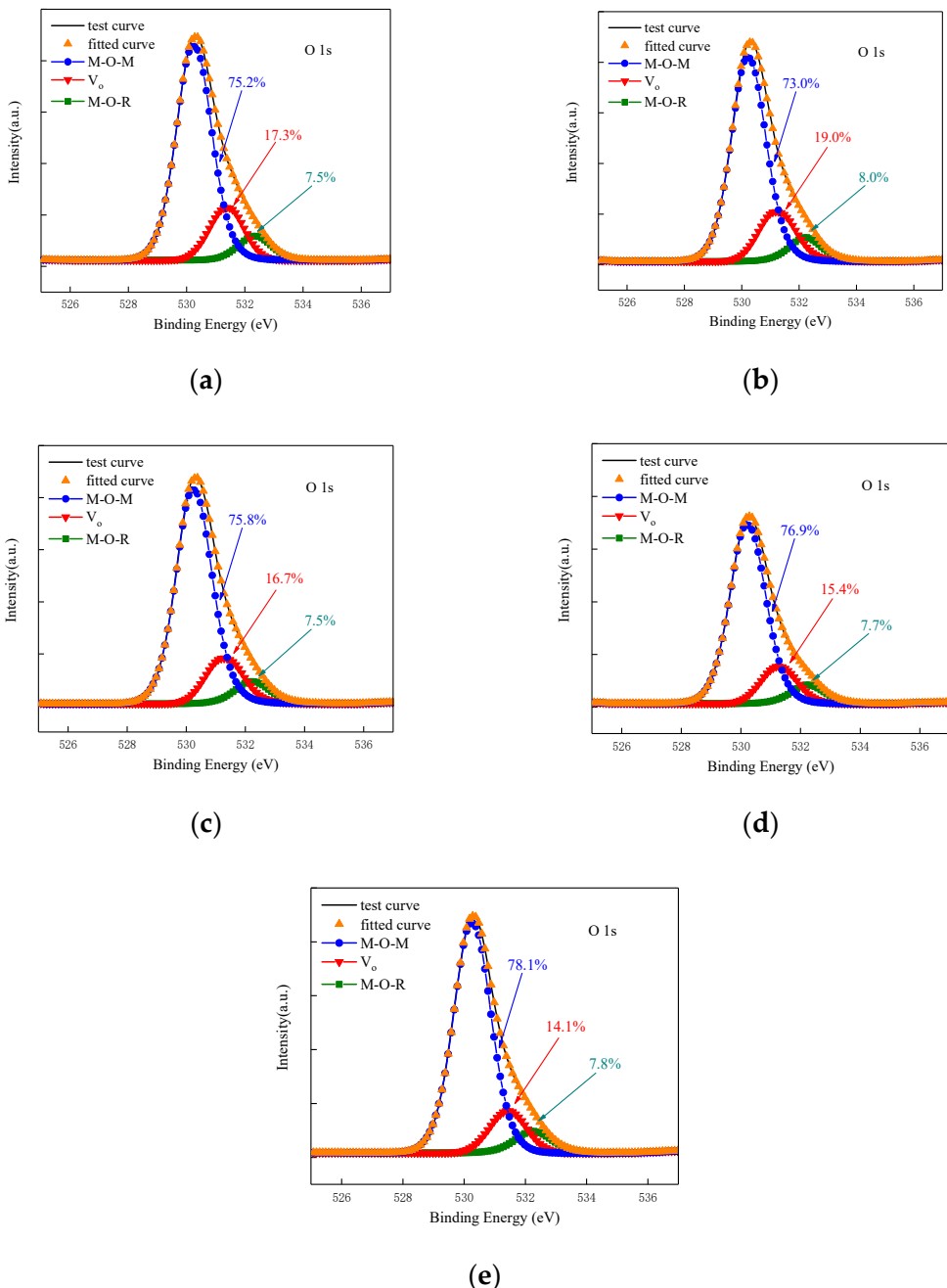

**Figure 3.** O 1 s XPS spectrum of Zr-AZTO films: (**a**) Sample A; (**b**) Sample B; (**c**) Sample C; (**d**) Sample D; (**e**) Sample E.

Figure 4 shows the transmittance of the Zr-AZTO thin films of different groups measured by UV-2600. At 600 nm, the transparency of Sample A~E is respectively 81.8%, 82.1%, 82.4%, 81.3%, and 80.6%. It shows that the transparency of Sample A is in the middle among five groups. The transparency of Sample B and Sample C is higher than the transparency of Sample A while the transparency of Sample D and Sample E is lower than the transparency of Sample A. Sample C and Sample E achieve the highest and the lowest transparency, respectively. When the wavelength is lower than 400 nm, the transparency of all samples declines rapidly.

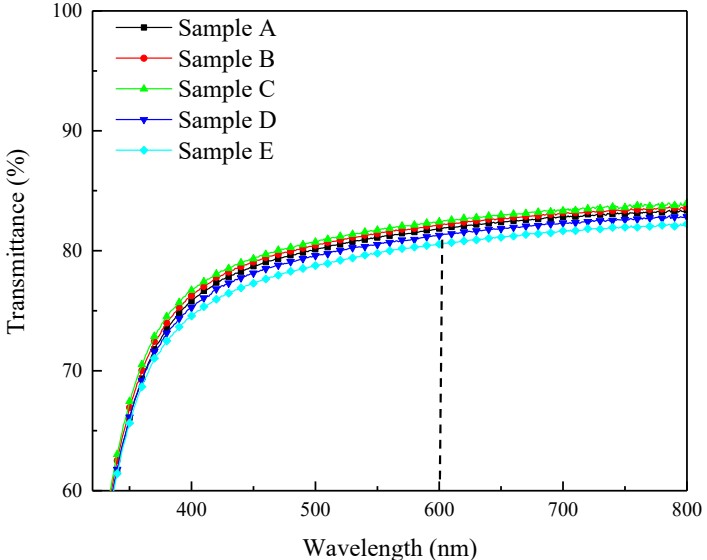

**Figure 4.** Transmission of Zr-AZTO thin films of different groups.

The optical band gap (Eg) of Zr-AZTO films can be calculated by Tauc's relation [20]:

$$(ahv)^2 = K(hv - E_g),\tag{1}$$

$$a = \frac{\lg\left(\frac{1}{T}\right)}{d},\tag{2}$$

where $a$ is the absorption coefficient, $K$ is constant, $hv$ is the photon energy, $T$ is the normalized transmittance, and $d$ is the thickness of films. Figure 5 is the plot of $(ahv)^{0.5}$ versus $hv$ for Zr-AZTO films of different power combinations [21]. With the reverse extension line from the linear part of curve intersecting with the horizontal axis, the value of optical band gap can be obtained at the intersection point. As shown in Figure 5, with the discharge power of AZTO increasing from 100 W to 120 W, the value of optical band gap decreases from 2.76 eV to 2.71 eV. With the discharge power of $ZrO_2$ increasing from 15 W to 45 W, the value of optical band gap is unchanged and remains at 2.86 eV.

Figure 6 shows the AFM (CSPM5500, Tapping mode) images of Zr-AZTO films of different power combinations annealed at 350 °C. From Figure 6, it is observed that the surfaces of all samples are flat. The RMS roughness of Sample A~E is respectively 0.446 nm, 0.402 nm, 0.387 nm, 0.399 nm, and 0.490 nm. For Sample A and Sample B, with the discharge power of AZTO target decreasing, the RMS roughness decreases as well. For Sample B~E, the RMS roughness decreases to 0.387 nm first and then increases to 0.490 nm as the discharge power of $ZrO_2$ target increases. It is found that the RMS roughness of different power combinations is approximate.

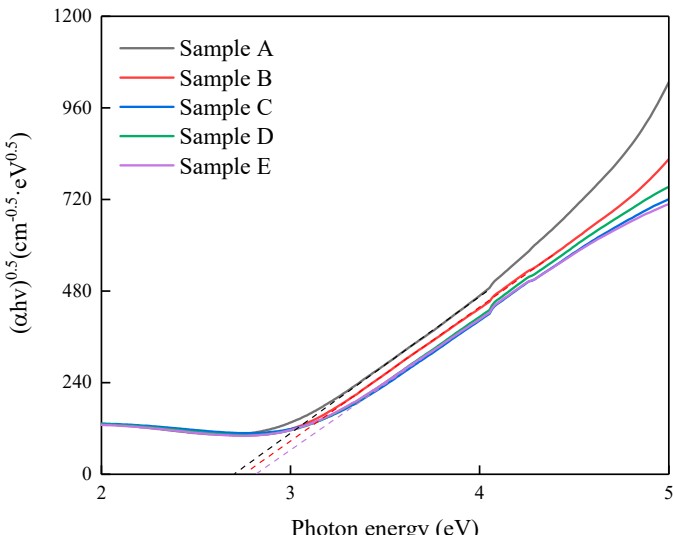

**Figure 5.** Plot of $(ahv)^{0.5}$ versus *hv* for Zr-AZTO films of different power combinations.

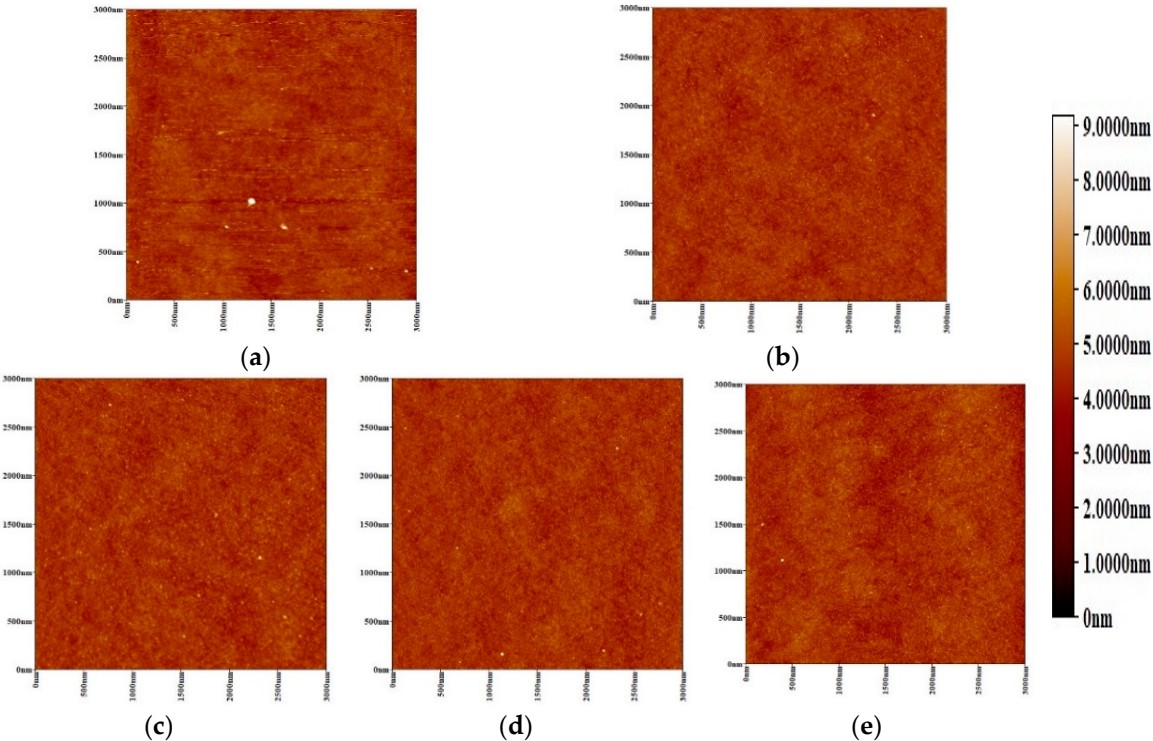

**Figure 6.** The AFM images of Zr-AZTO films with different Zr content annealed at 350 °C: (**a**) Sample A; (**b**) Sample B; (**c**) Sample C; (**d**) Sample D; (**e**) Sample E.

Figure 7 shows the crystalline of Zr-AZTO films of different power combinations annealed at 350 °C. There is no diffraction peak in the image after the films annealed at 350 °C. Thus, the Zr-AZTO films are amorphous in all conditions. This means in the film interior there is no anisotropy which can restrict the film applied on active layer. Moreover, the amorphous active layer can be used on large-scale display because of its good uniformity [22].

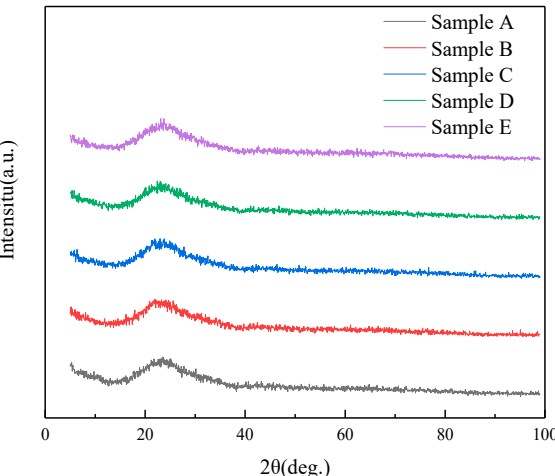

**Figure 7.** The XRD image of Zr-AZTO films of different power combinations annealed at 350 °C.

Figure 8 is the XRR images of Zr-AZTO thin films. By fitting with these curves, the physical parameters (like density, roughness and thickness) can be obtained. From Figure 8, it is found that the thin films of Sample A~E have similar curves, which indicates they have similar physical parameters. The physical parameters obtained from XRR curves are summarized in Table 3.

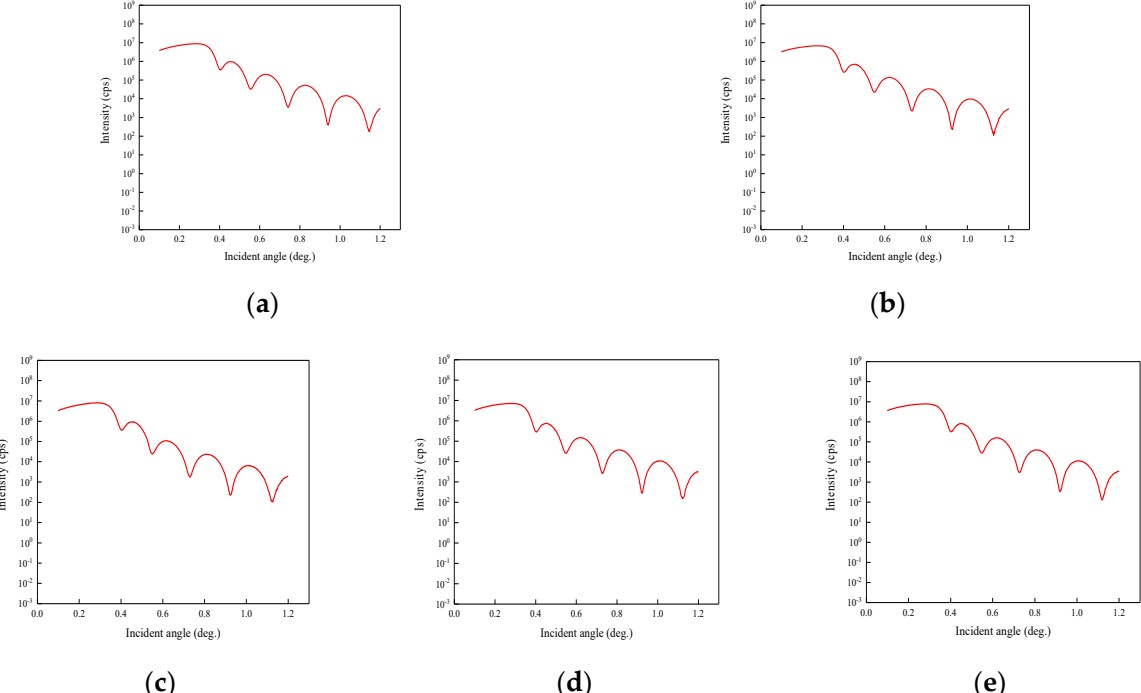

**Figure 8.** The XRR images of Zr-AZTO films: (**a**) Sample A; (**b**) Sample B; (**c**) Sample C; (**d**) Sample D; (**e**) Sample E.

Table 3 shows the physical parameters of Zr-AZTO films. Table 3 provides the density, thickness, roughness of Zr-AZTO films measured by XRR and other physical parameters mentioned in a previous part of this article. From Table 3, it is found that the Zr-AZTO films' density is around 5.90 g/cm$^3$, thickness is around 1.10 nm and thickness is around 20 nm. With the discharge power of AZTO target decreasing from 120 W to 100 W, the μ-PCD mean peak and D value respectively increase from 114.1 mV to 139.3 mV and from 1.24 to 1.54. With the discharge power of ZrO$_2$ target increasing from 15 W to 45 W, the mean peak and D value respectively decrease from 139.3 mV to 80.9 mV and from 1.54 to

0.77. As for the influence of the change of mean peak and D value, we will discuss these variables below after the electrical properties of devices are exhibited.

**Table 3.** The physical parameters of Zr-AZTO thin films.

|  | Sample A | Sample B | Sample C | Sample D | Sample E |
|---|---|---|---|---|---|
| Density by XRR (g/cm$^3$) | 5.97 ± 0.02 | 5.85 ± 0.12 | 5.84 ± 0.15 | 5.88 ± 0.09 | 5.90 ± 0.10 |
| Thickness by XRR (nm) | 20.2 ± 0.1 | 20.6 ± 0.1 | 20.6 ± 0.1 | 20.7 ± 0.1 | 20.8 ± 0.1 |
| Roughness by XRR(nm) | 0.96 ± 0.04 | 1.12 ± 0.03 | 1.16 ± 0.12 | 1.10 ± 0.10 | 1.09 ± 0.04 |
| Zr content (at.%) | 0.34 ± 0.01 | 0.63 ± 0.01 | 1.00 ± 0.01 | 1.78 ± 0.01 | 2.79 ± 0.01 |
| $S_{vo}/S_{all}$ (%) | 17.3 ± 0.8 | 19.0 ± 0.1 | 16.0 ± 0.7 | 15.4 ± 0.6 | 14.1 ± 0.1 |
| Transmittance at 600 nm (%) | 81.8 | 82.1 | 82.4 | 81.3 | 80.6 |
| RMS roughness (nm) | 0.446 | 0.402 | 0.387 | 0.399 | 0.490 |
| Optical band gap (eV) | 2.71 | 2.76 | 2.86 | 2.86 | 2.86 |
| μ-PCD mean peak (mV) | 114.1 | 139.3 | 110.8 | 95.8 | 80.9 |
| μ-PCD D value | 1.24 | 1.54 | 1.35 | 0.77 | 0.77 |

Figure 9 shows the output and transfer characteristics of Zr-AZTO TFTs of different power combinations. The electrical properties were measured by the semiconductor parameter analyzer (Agilent 4155C). As shown in Figure 9a–e, after a 350 °C annealing treatment, the devices of Sample A and Sample B have a similar on-state current while the devices of Sample C and Sample D have a similar but lower one. For the devices of Sample E, the on-state current is the lowest. For the transfer characteristics, Figure 9f shows that when the Zr content increases, the curves shift to positive direction. Because of the low standard electrode potential of Zr (−1.45 V), Zr can suppress the carrier concentration [5]. This result also reached a consensus with the XPS measurement. It is known that the oxygen vacancy can represent the carrier concentration [23]. When combined in Figure 3, the films of Sample A and Sample B have a similar carrier concentration, so their on-state current and $V_{on}$ are similar. The films of Sample C and Sample D have a lower oxygen vacancy concentration, which means they have a lower carrier concentration. As a result, their on-state current is lower and the transfer curves shift to a positive direction.

Table 4 shows the electrical parameters of Zr-AZTO TFT devices. The $\mu_{sat}$ means the carriers mobility in a saturation regime, and it can be calculated by the following equation:

$$I_{DS} = \mu_{sat} C_i \frac{W}{2L} (V_{GS} - V_{th})^2, \tag{3}$$

where $I_{DS}$ is the drain-source current, $C_i$ is the gate capacitance per unit area, $\frac{W}{L}$ is the channel width/length, $V_{GS}$ is the gate voltage, and $V_{th}$ is the threshold voltage. The SS is defined as the inverse of the maximum slope of the transfer curve:

$$SS = \left( \frac{d \log(I_{DS})}{dV_{GS}} |_{max} \right)^{-1}. \tag{4}$$

For an oxide semiconductor, there are two recombination processes: shallow localized states and deep localized states. The shallow localized states can be evaluated by the D value and the deep localized states can be evaluated by the mean peak. The high peak level and the high D value respectively represent the low density of deep localized states and the low density of shallow localized states [24,25]. It is known that the localized states could impose restrictions on the mobility of the semiconductor. Therefore, the low density of localized states can be conducive to the high mobility. Combining Table 3 with Table 4, it is found that the devices of Sample B can get the highest mean peak and D value, which means the density of deep localized states and shallow localized states reaches the lowest level at the same time. Correspondingly, the saturation mobility of Sample B is the highest level available.

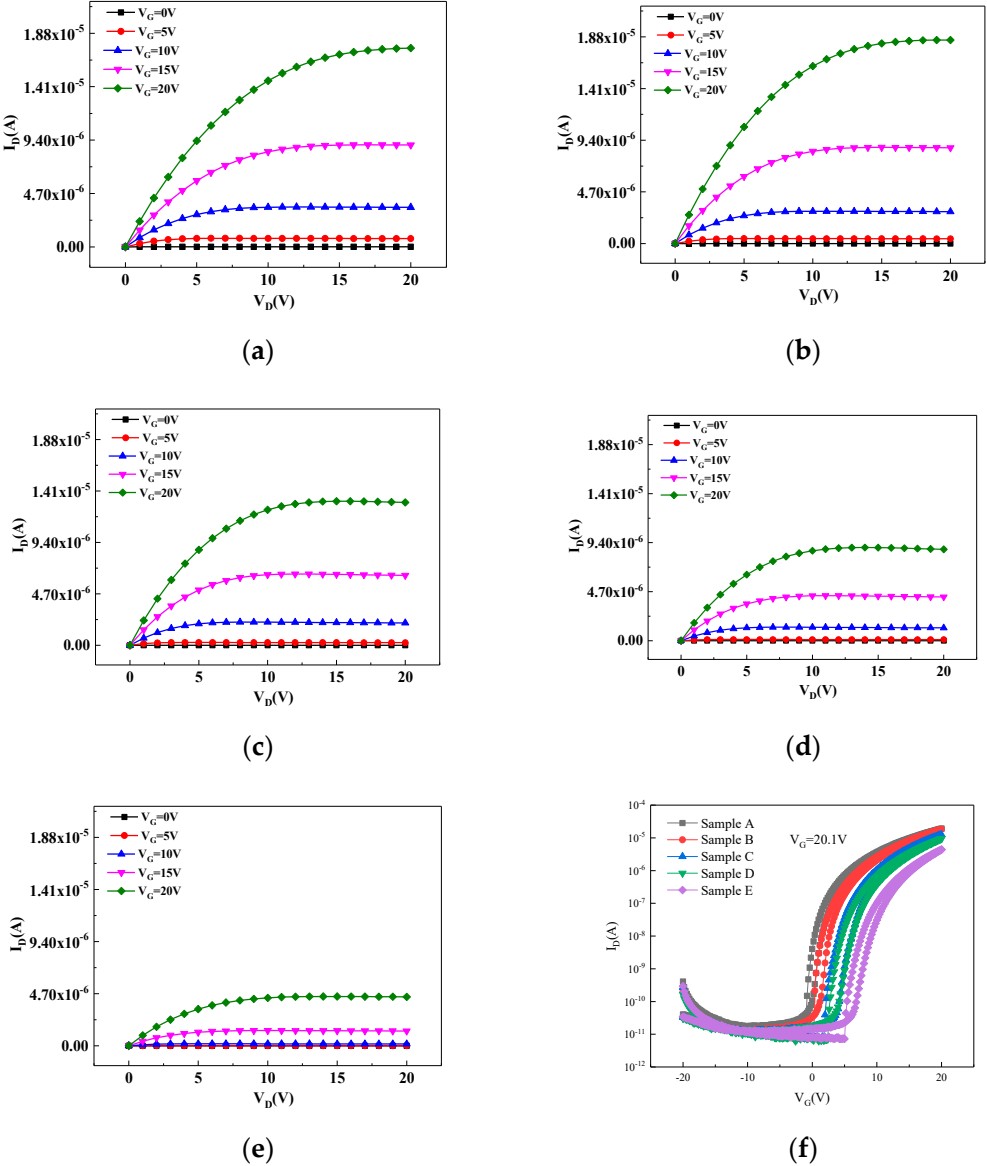

**Figure 9.** The electrical characteristics of Zr-AZTO thin film transistors the devices annealed at 350 °C. (**a**–**e**) The output curves of Sample A~E; (**f**) The transfer curve of Sample A~E.

**Table 4.** The characteristics of Zr-AZTO TFT devices annealed at 350 °C.

|  | Sample A | Sample B | Sample C | Sample D | Sample E |
|---|---|---|---|---|---|
| $\mu_{sat}$ (cm$^2$/(V·S)) | 7.1 ± 0.1 | 8.0 ± 0.6 | 7.6 ± 0.6 | 7.3 ± 0.3 | 5.2 ± 0.6 |
| Ion/Ioff | $(1.86 \pm 0.20) \times 10^6$ | $(2.01 \pm 0.34) \times 10^6$ | $(1.66 \pm 0.16) \times 10^6$ | $(1.45 \pm 0.27) \times 10^6$ | $(8.89 \pm 2.77) \times 10^5$ |
| SS (V/dec) | 0.23 ± 0.02 | 0.18 ± 0.03 | 0.25 ± 0.05 | 0.29 ± 0.05 | 0.27 ± 0.02 |
| Von (V) | −0.22 ± 0.12 | 1.08 ± 0.20 | 2.41 ± 0.14 | 3.04 ± 0.18 | 5.45 ± 0.11 |

As seen through comprehensive consideration, Sample B has a better electrical performance than other samples. Thus Sample B was chosen to measure the bias stress. Figure 10 shows the negative bias stress (NBS) and positive bias stress (PBS) of Sample B. The $V_G$ was applied at −20 V and 20 V for 0 s, 100 s, 600 s, 1200 s, 2400 s, and 3600 s. As Figure 10 shows, Sample B shifts 0.5 V to the negative direction after NBS was applied for 3600 s while it shifted 9.6 V to the positive direction after PBS was applied for 3600 s.

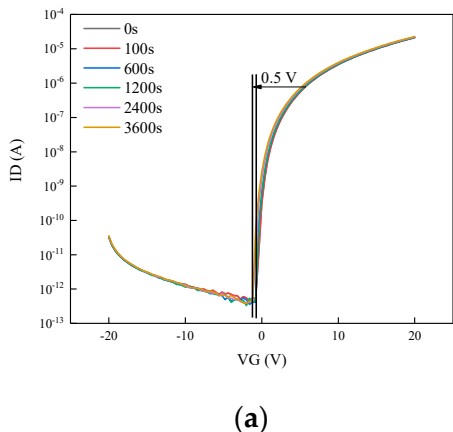
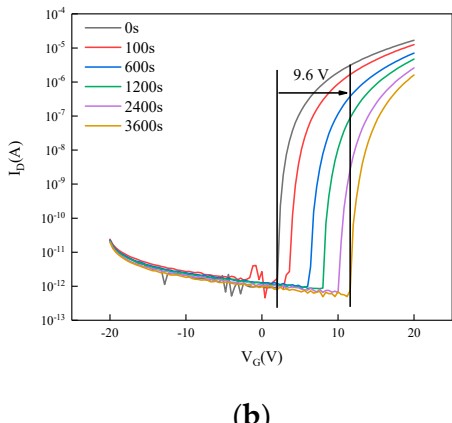

(**a**)　　　　　　　　　　　(**b**)

**Figure 10.** Transfer characteristic curves of Sample B as a function of the applied stress time (0, 100, 600, 1200, 2400, and 3600 s): (**a**) TFT device under NBS; (**b**) TFT device under PBS.

## 4. Conclusions

In conclusion, the Zr-AZTO TFTs of different co-sputter power were fabricated via the co-sputtering and annealed at 350 °C in amorphous air, while the thickness of active layer was controlled at 20 nm. The Zr content of Sample A~E was 0.34 ± 0.01 at.%, 0.63 ± 0.01 at.%, 1.00 ± 0.01 at.%, 1.78 ± 0.01 at.%, and 2.79 ± 0.01 at.%. The films of different power combinations had some similar physical properties, like a density of around 5.90 g/cm$^3$, thickness of around 1.10 nm and transmittance of around 81.6%. After being calculated by Tauc's relation, it was found that the optical band gaps of Sample A~E were in the range 2.71 eV–2.86 eV. After annealing treatment, the films remain amorphous, which means the TFT with Zr-AZTO as an active layer has good uniformity. Due to the low standard electrode potential, Zr can suppress the carrier concentration to some extent. Therefore, in Figure 8b, with the increase of the Zr content, the on-state current decreases and the transfer curves shift to the positive direction. The O 1 s and Zr 3d spectrum measured by XPS can also indicate that the Vo percentage decreases as the Zr content increases. The μ-PCD was used to evaluate the localized states of Zr-AZTO active layer. For Sample B, the active layer got a highest carrier density and the lowest localized states. Correspondingly, the devices of Sample B reached a performance with a μ$_{sat}$ of 8.0 ± 0.6 cm$^2$/(V·S), an Ion/Ioff of (2.01 ± 0.34) × 10$^6$, and a SS of 0.18 ± 0.03 V/dec at 350 °C. The device of Sample B had a 0.5 V negative shift under −20 V NBS and a 9.6 V positive shift under 20 V PBS.

**Author Contributions:** Conceptualization: R.Y. and H.N.; funding acquisition: H.N.; investigation: X.Z. and X.L.; data curation: X.Z., M.S. and Z.L.; supervision: Y.W. and J.P.; writing—original draft preparation: X.Z.; writing—review and editing: X.Z., K.L. and H.N.; validation: X.Z. and K.L.; visualization: D.G. and Y.W.; project administration: H.N. and R.Y.; funding acquisition: Y.Y.

**Funding:** This research received no external funding.

**Acknowledgments:** This work was supported by National Key R&D Program of China (No. 2016YFB0401504), National Natural Science Foundation of China (Grant. 51771074), Guangdong Science and Technology Project (No. 2016B090907001) and the Fundamental Research Funds for the Central Universities.

**Conflicts of Interest:** The authors declare no conflict of interest.

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
