# Peer review of "The Performance of Zr-Doped Al-Zn-Sn-O Thin Film Transistor Prepared by Co-Sputtering"

_applsci, doi:10.3390/app9235150_

Round 1

Reviewer 1 Report

This work reports on the performance of Zr-doped AZTO transistors prepared by co-sputtering. The work is in principle interesting but the presentation can be improved.

First of all there are minor English-language problems, e.g. on page 5, line 120: 'There is no any diffraction peaks...' should better read: 'There is no diffraction peak...'. The authors should make a careful correction of all such aspects.

Page 2, line 60: The annealing temperature of 350 does not damage the devices due to interdiffusion?

I do not see discussion of error bars in this work, while many parameter values change negligibly. Error bars should be provided and discussed throughout the paper. 

I would like to see more details on the AFM measurements. What AFM instrument was used, which operation mode, tips and tip radius. Moreover, some surface plots instead of only 3D plots should be provided.

Author Response

Dear Reviewer 1:

We really appreciate the valuable and professional suggestions for our manuscript entitled “The performance of Zr-doped Al-Zn-Sn-O thin film transistor prepared by co-sputtering”. We have carefully addressed our manuscript based on the reviewers’ comments, and the changes to the original manuscript are highlighted in the revised version by using the "Track Changes" function in Microsoft Word. Below please check for our response to the comments, which we reflect in our revised manuscript.

Please kindly let us know if you have any questions.

With best regards

Xiaochen Zhang

South China University of Technology

Reviewer 2 Report

In the manuscript authors deal with thin film transistor based on Zr-doped aluminium-zinc-tin oxide (AZTO) thin film deposited by RF co-sputtering (ZrO2 and AZTO target). They have analysed the influence of discharge power for both targets on chemical content, structure and electrical properties of deposited thin films performance of complete TFT device.

The manuscript is well structured in general but there are several parts in Introduction, Materials and Methods and especially Results and Discussion that should be clarified.

Introduction chapter is without state of the art related to Zr-doped AZTO, what have been done until now with this material (preparation, properties, comparison with other similar materials) and also about the application in TFT. Also main aim, contribution and novelty of this work (Zr-doped AZTO TFT performance optimization by sputtering power variation) were not clearly stated.

In Materials and Methods are missing some details about sample preparation, which permits readers to reproduce presented experiments and results. For Zr-doped AZTO layer deposition parameters are partially given (for example target size, substrate to target distance, deposition temperature). Also for Al-Nd alloy layer deposition, patterning by wet etching, gate electrode deposition by anodic oxidation. This is not central contribution of this work but should be given.

I don’t understand the result of postdeposition annealing process at 350 C deg. because samples stays amorphous after annealing. What is a function of annealing process? How the structure of as-deposited samples looks like compared to XRD presented in Figure 6.

Authors discussed variation in samples properties (chemical content, surface roughness, transmission) with sputtering discharge power but without errors for presented parameters numerical values. It is difficult to estimate statistical significance of this variations. I’m suggesting to add error to results for all presented results: atomic content values, surface roughness, optical gap, etc.

Section about AFM roughness discussion needs several explanations. What does it mean “relatively flat”? Compared to what? Also, values for RMS roughness with precision to 0.001nm and related discussion do not make sense. An instrument used for AFM is not specified but in general vertical precision/resolution of nowadays instruments is in angstroms range (especially because of AFM tip geometry and dimensions constraints), so I would say that there is no significant difference in roughness for samples A-E. I’m not convinced that it’s possible to discuss AZTO samples morphology at atom levels based on AFM images.

I’m suggesting to add in Table 2 all numerical results mentioned in the text for transmittance @600nm, XPS Zr and O2 results, optical gap, roughness. It would be easier to compare and discuss results variations.

For XRR results I’m suggesting to add Figure of XRR as a function of incident angle for all. From XRR results can be calculated also surface roughness by using a proper model. It would be nice to compare results obtained In this way with AFM roughness.

Regarding electrical measurements results presented in Fig 8 should be clarified in more details. First, in the text are mentioned four subfigures a), b), c) and d) of Fig 8 but are presented only 2. Also, there are only 5 samples but much more curves at Fig. 8a. This should be clarified. At Fig 8b there are lot overlapping curves. Also I’m suggesting to compare results with other TFT standard materials from literature in the discussion.

In my opinion, after major revision, the manuscript can be considered again for publication in the “Applied Science” journal.

Other minor corrections and comments that can improve the manuscript:

there are several abbreviations that are used but not defined. For example “XPS” at page row 40, micro-PCD,  page 2, row 46. (I know that for someone familiar with that field this is not a problem).

page 2, row 41: the term “power of ZrO2” better to use discharge power of

page 2, row 43: the term “high oxygen vacancies” is not clear. Mean concentration/number of vacancies?

Page 2, row 45: I don’t understand the statement about amorphous nature of AZTO layer and uniformity. Amorphous layer as well as crystalline layer can be non-uniform? At least should be supported by reference.

Page 2, row 46: the term “mobility of TFTs” not clear. Charge carrier mobility?

Page 2, row 47: the term “low localized states” not clear. Concentration/number or something else?

Page 2, Materials and Methods: How the layers thickness is measured? XRR is used for all layers (Nd:Al, AlO)

Page 2, Materials and Methods: Which instruments were used AFM, micro-PCD, etc?

Page 5AFM roughness is RMS roughness? Should be specified.

XPS results and discussion: I’m suggesting to put all results and discussion regarding XPS at one place (Fig. 2 and Fig. 7). What about other elements: Al, Ti?

Fig 1. To get an impression about the TFT sample size I’m suggesting to add rough dimensions of the device to Fig or somewhere in the text.

Fig 3. Inset without scales (horizontal and vertical) doesn’t provide any new information about transmittance variation. I’m suggesting to add axis. Also, it’s not clear is the presented transmittance for only AZTO layer or for AZTO + substrate.

Fig 4. Similar comment as for Fig. 3. How the absorption coefficient used in Fig 4 is calculated from transmittance? At least reference should be given? Which energy range was used for gap calculation by linear fit?

Fig 6. What XRD geometry was used to obtain presented diffractograms (an instrument used and wavelength not specified). Usually for thin films is used grazing incidence geometry. In angular axis label should be used Greek letter.

At page 7, row 154: Statement “the oxygen vacancy can represent the carrier concentration” should be clarified or at least supported by the reference.

Table 3: Parameters in the first column should be defined and described how they are evaluated/calculated. At least reference should be given.

Author Response

Dear Reviewer 2:

We really appreciate the valuable and professional suggestions for our manuscript entitled “The performance of Zr-doped Al-Zn-Sn-O thin film transistor prepared by co-sputtering”. We have carefully addressed our manuscript based on the reviewers’ comments, and the changes to the original manuscript are highlighted in the revised version by using the "Track Changes" function in Microsoft Word. Below please check for our response to the comments, which we reflect in our revised manuscript.

Please kindly let us know if you have any questions.

With best regards

Xiaochen Zhang

South China University of Technology

Reviewer 3 Report

In this work, authors have investigated the Zr-doped aluminum-zinc-tin oxide (Zr-AZTO) TFTs. Authors well compared the effect of process parameter on TFT performances and the reason of such a difference. However, I think that many studies on the correlation of sputtering parameters with TFT performances have been already reported over several decades. Recently, the device reliability has been a key issue for next-generation displays. As authors know, IGZO TFTs are already used in OLED TVs. Thus, I feel that the originality and novelty of this work is insufficient for the publication. Therefore, I suggest that authors should add the stability data such as bias stress or illumination stress. Furthermore, before this work is publishable, the authors should respond to the following concerns.

In Figure 4, authors used the equation of (ahv)2 to determine the optical bandgap of AZTO. However, for amorphous oxide semiconductors, the relation of (ahv)0.5 should be used. See the following papers. (J. Kim et al, NPG Asia Materials. 9.3 (2017) :e359, J. Kim et al, Thin Solid Films 614 (2016) 84-89). For XPS analyses, authors should summarize the table that indicates the details such as FWHM and binding energy. Moreover, authors should confirm again that the FWHMs for M-O, Vo, M-OH are the same or not. If the FWHMs are different for each fitting, which means that the fitting results are not reliable. If the proposed Zr-doped AZTO has a wide-band gap compared to conventional IGZO, the illumination stress must be better than the IGZO TFTs. See the following paper. (J. Kim et al, APL Materials, 7 (2019) 022501)

Author Response

Dear Reviewer 3:

We really appreciate the valuable and professional suggestions for our manuscript entitled “The performance of Zr-doped Al-Zn-Sn-O thin film transistor prepared by co-sputtering”. We have carefully addressed our manuscript based on the reviewers’ comments, and the changes to the original manuscript are highlighted in the revised version by using the "Track Changes" function in Microsoft Word. Below please check for our response to the comments, which we reflect in our revised manuscript.

Please kindly let us know if you have any questions.

With best regards

Xiaochen Zhang

South China University of Technology

Round 2

Reviewer 1 Report

This version is improved. The only technical point I see and can be improved is the very small labels in new afm Fig. 6.

Author Response

Dear editor Wang,

     We really appreciate the valuable and professional suggestions by you and the reviewers for our manuscript entitled “The performance of Zr-doped Al-Zn-Sn-O thin film transistor prepared by co-sputtering”. We have carefully addressed our manuscript based on the reviewers’ comments, and the changes to the original manuscript are highlighted in the revised version by using the "Track Changes" function in Microsoft Word. Below please check for our response to the reviewers’ comments point-by-point, which we reflect in our revised manuscript.

Please kindly let us know if you have any questions.

With best regards

Xiaochen Zhang

Reviewer 2 Report

The authors have taken the reviewers' comments seriously and revised the manuscript accordingly. In my opinion, the revised manuscript could be considered for publication in the Coatings journal. However, there are still some typing errors but I assume that that can be corrected during the next step.

Author Response

(The authors gave the same response as above.)

Reviewer 3 Report

 Authors have well responsed to my questions and revised the manuscript. I suggest the publication of this manuscript in the Applied Sciences.

Author Response

(The authors gave the same response as above.)
